# Sentiment Classification of Historical Danish and Norwegian Literary Texts

**Ali Al-Laith[1,2], Kirstine Nielsen Degn[1], Alexander Conroy[1],**
**Bolette Sandford Pedersen[1], Jens Bjerring-Hansen[1] and Daniel Hershcovich[2]**
Department of Nordic Studies and Linguistics, University of Copenhagen[1]
Department of Computer Science, University of Copenhagen[2]
alal@di.ku.dk, kirstinenielsendegn@gmail.com, alc@hum.ku.dk,
bspedersen@hum.ku.dk, jbh@hum.ku.dk, dh@di.ku.dk

## Abstract

Sentiment classification is valuable for literary analysis, as sentiment is crucial in literary narratives. It can, for example, be used to investigate a hypothesis in the literary analysis of 19[th]-century Scandinavian novels that the writing of female authors in this period was characterized by negative sentiment, as this paper shows. In order to enable a data-driven analysis of this hypothesis, we create a manually annotated dataset of sentence-level sentiment annotations for novels from this period and use it to train and evaluate various sentiment classification methods. We find that pre-trained multilingual language models outperform models trained on modern Danish, as well as classifiers based on lexical resources. Finally, in the classifier-assisted corpus analysis, we both confirm and contest the literary hypothesis and further shed light on the temporal development of the trend. Our dataset and trained models will be useful for future analysis of historical Danish and Norwegian literary texts.

## 1 Introduction

Sentiment analysis, the computational study of emotions, opinions, and evaluations expressed in text, has become an important tool in natural language processing. It is based on the premise that words are associated with sentiments or valence, and that these associations can be quantified (Thomsen et al., 2021). However, its application in literary studies has been limited. Literary texts, such as novels, poems, and plays, provide a unique cultural window into past attitudes, beliefs, and emotions. By analyzing the sentiment expressed in these texts, researchers can gain a deeper understanding of the cultural and societal attitudes of the past and how they have shaped our present understanding of the world. However, traditional sentiment analysis techniques in the investigation of literary texts may be less effective due to the use of an archaic vocabulary, literary ambiguity, and figurative language, as well as the limited training data, the difficulties of generalizing models trained on modern text, and the challenges of annotation. Therefore, there is a need for new methods and approaches to analyze sentiment in historical and literary texts to uncover the valuable insights they can provide. However, there is no existing benchmark for sentiment analysis of the texts we are interested in, and it is not known how well existing, general methods perform on them, and whether they can be used for a meaningful analysis of literary hypotheses.

In this paper, we (1) introduce an annotated sentiment dataset of historical literary Danish and Norwegian, (2) evaluate various classification models on the dataset, and (3) use an accurate classifier to analyze a large historical literary corpus and provide initial evidence for a literary hypothesis as outlined in the following section. Our code and data are available at https://github.com/mime-memo/unhappy.

## 2 Literary Context: The Unhappy Texts

In our experiments, we focus on literature from the so-called 'Modern Breakthrough,' which denotes the literary currents embedded in the new social realist and naturalist literature which blossomed in Scandinavia during the 1870s and the 1880s as well as a cultural and societal transformation, encompassing politics, morals, gender roles, etc (Ahlström, 1947). On the whole, the historiography of the breakthrough, largely concentrated on a few canonical male authors, does not reflect the diversity of the period's literary production, especially in terms of gender. Women were most noticeable among the new groups of authors stepping

forward from 1870-1900. As a rule, female authors were not recognized by contemporary (male) criticism; the famous critic Georg Brandes introduced the term breakthrough in a book grouping only male Scandinavian authors, *The Men of the Modern Breakthrough* (Brandes, 1883). and, a generation later, the influential literary historian Vilhelm Andersen dubbed the period's female literary production as an 'odd province' of literature (Andersen, 1925). While Andersen recognized the period's literary production by Danish female authors, a much later literary handbook *Hovedsporet* (Jørgensen, 2005) does not. Its chapter on the debates on gender in the period's literature has the heading 'Women's literature, written by men'.

In fact, for many decades of the 20th century, the female breakthrough authors were overlooked and mostly left out of literary histories, but in the 1970's and 80's, as a part of a greater feminist wave in Scandinavian literary studies, scholars made a great effort to reinterpret the period. A result of this was the prevalent hypothesis about the works of the female authorships from the period being characterized as 'unhappy texts' (Dalager and Mai, 1982; Jensen et al., 1993). The argument is unfolded in greatest nuance in Pil Dahlerup's doctoral dissertation, *The women of the modern breakthrough* (Dahlerup, 1984). Dahlerup argues that because the women of the period were restrained by the patriarchal society, they also wrote fictional characters who lacked agency, were unhappy and disillusioned. Meanwhile the scholars highlight that the emancipated female characters of the period are first and foremost present in the works of male authors, arguing that the male authors, who enjoyed both private and public liberty, had the capacity to portray fictional characters with the same liberties (Dahlerup, 1984).

Our investigation will revisit this hypothesis with two points of departure, one regarding empirical selection and scale, and another regarding methodology. The hypothesis of the unhappy texts is based solely on texts by female authors and often on relatively few texts. The first point of departure is thus to revisit the hypothesis with a quantitative perspective, also including the works authored by men. Our second point of departure, in revisiting the hypothesis of the unhappy texts, is based on a theoretical framework from the field of gender studies and feminist literary studies. Central to our inquiry are new insights from affect

theory and the notion of sentiment and affects as something humans 'do' rather than an underlying *a priori* structure as is the case in the psychoanalytic assumptions that the hypothesis of the unhappy texts is based on. We thus set out to find new methodologies to test the hypothesis of unhappy texts. In this experimental methodological phase, we find that sentiment analysis, analyzing the valence associated with a given word, sentence, or text, is a meaningful starting point because of its juxtaposition of affect theoretical and quantitative perspectives.

# 3 Related Work

In this section, we present sentiment-related works in both literature and historical corpora.

## 3.1 Sentiment Analysis on Literature

The study of emotions in literature has become an integral part of literary analysis with the emergence of digital humanities. This field of research focuses on using computational methods to understand emotions in literature. This encompasses a wide range of topics, from tracking changes in the plot to analyzing the emotional content of texts (Kim and Klinger, 2019).

Emotion and sentiment classification involves classifying text into predefined classes based on emotions/sentiment. This task is applied in the literature to group literary texts based on their emotional properties. Some studies have focused on classifying the emotions in works of Francisco de Quevedo's poems (Barros et al., 2013), American poetry (Reed, 2018), and early American novels (Yu, 2008). Volkova et al. (2010) annotated in fairy tales, while Ashok et al. (2013) used sentiment polarity to predict the success of a book. Zehe et al. (2016) used sentiment classification to classify 212 German novels into happy or non-happy endings. They showed encouraging results using support vector machines. While some studies approach the task as a classification problem, others focus on the structural changes of sentiment and emotions. Heuser et al. (2016) explore the relationship between emotions and geographical locations and Taboada et al. (2006) track sentiment and emotions towards certain groups.

## 3.2 Sentiment Analysis on Historical Text

Identifying and tracking the sentiment of text over time is challenging due to language variation, the

dynamic nature of sentiment, and the scarcity of historical corpora. Sentiment can change depending on a variety of factors, such as context, culture, and time. Additionally, historical and diachronic data may have different characteristics than contemporary data, such as changes in language use and writing styles. Along with the above challenges, the main focus of research nowadays is on temporal corpora in the news (Souma et al., 2019), and social media (Hazimeh et al., 2019), while little attention has been devoted to the historical domain (Sprugnoli et al., 2016).

Several techniques are used for sentiment analysis on diachronic and historical corpora. Schmidt and Burghardt (2018) used sentiment analysis on a German drama text corpus and evaluated the performance of different German sentiment lexicons using a manually annotated gold standard of 200 speeches. This study created an annotated corpus for the sentiment analysis of historical texts and revealed key issues related to the annotation and pre-processing of historical texts. Sprugnoli et al. (2016) analyzed an Italian corpus of writings of Alcide De Gasperi and developed a new lexical resource for sentiment analysis. The study found that crowd-sourcing was more effective for sentiment analysis of historical texts than using a sentiment lexicon. Hills et al. (2019) analyzed national subjective well-being using millions of digitized books in six different languages and countries. They found that Gross Domestic Product (GDP) and life expectancy have a strong positive effect on well-being, while conflict has a negative effect. These studies demonstrate the challenges and potential of sentiment analysis of historical texts and the importance of manual annotation and crowd-sourcing for accurate analysis.

Schmidt et al. (2021) analyzed emotional expression in historical German plays using various methods, including lexicon-based, traditional machine learning, word embeddings, and pre-trained and fine-tuned language models. The latter achieved state-of-the-art results, while lexicon-based and traditional machine learning consistently outperformed. However, performance decreases significantly with multiple sub-emotions.

To summarize, the challenges in sentiment analysis of historical texts, including lack of native speakers, limited data, unusual textual genres, and historical language, call for innovative and robust methods (Sprugnoli, 2021). Pre-trained language models can address these challenges by being trained on large corpora, fine-tuned on small datasets, and able to identify sentiment in diverse texts (§5).

|  | Main Corpus | Sub-corpus |
|---|---|---|
| Total novels | 839 |  |
| Total sentences | 3,229,137 | 2,748 |
| Total words | 52,724,457 | 55,333 |
| Average sentences per novel | 3,849 |  |
| Average words per novel | 62,842 |  |
| Average words per sentence | 16 | 20 |

Table 1: Corpus statistics for the main corpus and sub-corpus used in sentiment analysis.

## 4 Dataset of Historical Literary Text

This section describes our main corpus of historical literary text and a sub-corpus annotated for sentiment.

### 4.1 Main Corpus

We rely on the MEMO corpus (Bjerring-Hansen et al., 2022), comprising 839 Danish and Norwegian novels spanning the last 30 years of the 19th century and including more than 50 million words in total. We refer to this corpus as the 'main corpus'. The corpus is a rich and diverse collection of texts that will provide valuable insights into the registered sentiments and emotions of the period under investigation. Table 1 shows statistical information about the corpus.

### 4.2 Annotated Sub-Corpus

To ensure the accuracy and reliability of our sentiment classifiers, we systematically annotated a representative subset of sentences from our main corpus. We carefully selected 2,748 sentences, averaging 3.3 sentences per novel, to create a diverse sample. To minimize bias, we employed random sampling by shuffling the sentences and selecting 3-4 sentences from each novel. This method ensured that the annotations accurately reflect the sentiment distribution in the entirety of the corpus and laid a strong foundation for further research in sentiment analysis of historical novels. Additionally, Table 1 provides further statistical information regarding the sub-corpus.

### 4.3 Annotation Process

The annotation was conducted by three trained literary scholars: a master's student, a Ph.D. stu-

dent, and an associate professor. All three are native Danish speakers and annotated 690, 1029, and 1029 text segments from the corpus, respectively. The annotators shared domain knowledge in 19th century Scandinavian literature, with overarching and intertwining areas of expertise, but also particular interests (gender, history of ideas, and cultural history).

**Guidelines.** With respect to the principle that clear and simple instructions are crucial for obtaining high-quality annotations (Mohammad, 2016), and acknowledging the array of intricacies and bafflements, which an emotional analysis of literary texts based on small fragmentary segments raises, the guidelines were rigorously minimalist and pragmatic. Our guidelines are a simple sentiment annotation questionnaire with clarifying annotation directions.

1. The text segments were to be labeled either 'Positive', 'Negative', or 'Neutral'. The annotator was to assess which of these labels was most descriptive of the overall sentiment expressed in the segment.

2. Only the segment in question should be considered. Contextualisation and 'guessing' on what might go on before or after the segment were ruled out. In cases of doubt, the label should be 'Neutral'.

3. Attention should be paid to the historical fluctuations in language and semantic change. As an example the adjective 'besynderlig', today means 'weird', while it in the 19th century also had the meaning 'special' or 'curious'. This means that the following sample should be labeled as positive, rather than, and contrary to an anachronistic reading, negative:

   > 'Hun blev strax noget fortumlet over disse uforberedte Kjærtegn ; men da hun laa i hans Arme , saa' hun op paa ham med et besynderligt, ikke fornærmet Blik' (English translation: 'She was at once somewhat taken aback by these unprepared caresses; but as she lay in his arms, she looked up at him with a curious, not offended, look')

4. Finally, pragmatism was to be deployed by the annotators. Since the segments are short and heterogeneous (containing both dialogue, non-dialogue/description, or a mixture), no extra weight could be given to particular word classes (such as verbs, adjectives, or nouns).

**Challenges.** Since sentiment analysis has traditionally been used on texts with a strong valence and subjectivity (e.g. reviews or tweets), literary texts such as novels pose a challenge because they often are characterized by ambiguity and can be understood and interpreted in several different ways. Therefore, a moderate inter-annotator agreement (IAA) is also to be expected (Schmidt et al., 2019).

Another fundamental challenge in the annotation of imaginative texts reflects a key question in narratology (i.e., the study of narrative structure in texts): Who is speaking? In this context, distinctions are made between the text's (implied) author, narrator, and characters as well as between 'mood' and 'voice' as levels in literary texts (Genette, 1983). Since the narratological structure is often unclear, the annotation cannot aim at deciding whether the dialogue or text voices reflect the author's sentiment. At the risk of not taking into account literary devices such as irony and unreliability, the annotation can only address the dominant sentiment in the segment.

In addition, particular attention was paid to the following issues:

*Lack of context.* Often, segments with clear indications of emotion are ambiguous or vague due to a lack of context. We decided to label such instances as 'Neutral', even though they in reality, might have formed parts of negative vis-à-vis positive discourses. Example (translated): 'When the letter, finally, was finished, she folded it and went with a beating heart to the nobleman's door'. Here the valence is clear (something emotional is going on), but the polarity is unclear (is it something good or bad?).

*(Unconscious) Contextualization.* We tried our best to rely on our judgments on close reading combined with our familiarity with historical modes of morality and sensibility, including standards of courtesy and decorum, as expressed in speech as well as gesture and action. However, it is difficult not to contextualize or to rule out the role of (unconscious) contextualization. For instance, our knowledge of individual authors or texts and their distinctive traits (e.g., the Norwegian author Amalie Skram, famous for her coarse and unsentimental naturalistic style, permeated with negativity) or paratextual effects such as the connotations

of a book title (e.g., the novel *En Krise* (1892), 'A Crisis' by Johanne Schjørring, inevitably giving the reader a negative vibe towards the text).

*Gender.* Finally, and crucially in this context, is the fact that cultural values and norms change over time. Awareness of the issue of cultural value change over time, affecting the annotation (gender roles, gendered behavior, then and now). Relying on fundamental insights from social constructivist gender theory, understanding gender as a category subject to change in form and meaning over time, special attention was paid to segments involving gendered behavior and dialogue. A few examples can illustrate how cultural change poses challenges to annotation. First, we have a segment highlighting a male character, whose dominant behavior would be interpreted more negatively today than in the past:

> 'aa ja , men ofte gjorde han sig med Vilje haard , ti saaledes, var det bedst for dem, han havde at gøre med: "du skal!"' (Translation: 'oh yes, but often he made himself tough on purpose, for this was the best for those he had to deal with: "you must!"')

In the thematization of female gender roles, a modern understanding could potentially come at odds with the intentions in 19$^{th}$ century texts. This is highlighted in the following segment presenting a female character:

> 'Men allerhelst laa hun dog i Vinteraftenernes Skumring i sin Yndlingsstilling i Armstolen og grublede og drømte og ventede og ventede — ligesom Prinsessen i Eventyret.' (Translation: 'But preferably she would lie in the twilight of the winter evenings in her favourite position in the armchair and pondering and dreaming and waiting and waiting — like the princess in the fairy-tale.'

Today, the character's passivity, dreamfulness, and nostalgia are more likely to be perceived as negatives than positives.

### 4.4 Annotation Results

The sentiment annotations for all sentiment classes are illustrated in Table 2, which displays the distribution of the samples and sentiment classes.

**Agreement.** We use Cohen's Kappa to determine IAA on 100 samples annotated by all three experts, resulting in a score of 0.59, which indicates moderate agreement among annotators. This is likely a result of the subjectivity of the task and the challenges encountered in determining sentiment with limited context (see Challenges above).

| Sentiment Class | Total Samples | Percentage |
|---|---|---|
| Negative | 1,139 | 41.4% |
| Neutral | 788 | 28.7% |
| Positive | 821 | 29.9% |
| **Total** | 2,748 | 100% |

Table 2: Distribution of sub-corpus samples and sentiment categories.

## 5 Experiments

The dataset is split into three sets: training, validation, and testing to facilitate the development and evaluation of our models. The training set comprises 2376 examples, accounting for approximately 86% of the dataset. The validation set used for hyperparameter selection consists of 272 samples, representing about 10% of the entire dataset. Lastly, the testing set, which is utilized to evaluate the final performance of the model, contains 100 examples, representing approximately 4% of the total dataset. In the case of the training and validation sets, annotations were performed by a single expert. However, for the testing set, samples were annotated by all three experts, and the final label was determined through a majority vote. We use F1-score as our evaluation metric.

### 5.1 Lexicons and Models

In this section, we outline the lexicons and models evaluated in our sentiment classification experiments using both lexical-based, supervised without fine-tuning, and supervised with fine-tuning methods. Importantly, all lexicons and models are based on modern Danish lexica and/or training data, with no exposure to historical Danish or Norwegian. It should be noted that, until 1907, written Norwegian was practically identical to written Danish (Vikør, 2022). In section 5.2, 5.3, and 5.4, we show detailed information on how these resources were employed to achieve sentiment classification results. The following provides a concise overview of the lexicons and BERT models employed in our experiments.

**Sentida.** A Danish lexicon[1] comprised of the existing Danish sentiment lexicon AFINN and a list of new words. It scores sentences based on their words and provides either an average sentiment

---

[1] https://github.com/Guscode/Sentida

score or a total score. Sentida takes into account adverb-modifiers, exclamation marks, and negations in its sentiment scoring process (Lauridsen et al., 2019).

**TextBlob.** A Python package[2] that utilizes a lexicon-based approach in which sentiment is determined by the semantic orientation and the strength of each word in the sentence, using a pre-existing English dictionary that categorizes words as negative, neutral, or positive. We employ a two-step process to ensure accurate sentiment in Danish text. First, we utilize Google Translate[3] to translate Danish text to English. Then, we feed the translated text into TextBlob.

**VADER.** VADER (Valence Aware Dictionary and sEntiment Reasoner) is a lexicon and rule-based sentiment analysis tool attuned explicitly to sentiments expressed in social media and works well on texts from other domains (Hutto and Gilbert, 2014).[4] To adjust the VADER sentiment analysis technique for Danish, we use a Danish sentiment lexicon (Nimb et al., 2022; Pedersen et al., 2021)[5] containing a list of words and their associated sentiment scores.

**Danish Model BotXO.** This BERT model was developed by Certainly (previously BotXO) (Devlin et al., 2019). It has been pre-trained on 1.6 billion Danish words and is freely available[6]. This particular model has not been fine-tuned on sentiment classification.

**Danish BERT Tone.** The BERT Tone model[7] was developed to detect sentiment polarity (positive, neutral or negative) in Danish texts. The model was constructed by fine-tuning the BotXO Danish BERT model.

**Danish Sentiment.** This model[8] is a fine-tuned version of the multilingual pre-trained model XLM-RoBERTa-base (Conneau et al., 2020). It

---

[2]https://textblob.readthedocs.io/
[3]https://translate.google.com/
[4]https://github.com/cjhutto/vaderSentiment
[5]https://github.com/dsldk/danish-sentiment-lexicon
[6]https://huggingface.co/Maltehb/danish-bert-botxo
[7]https://huggingface.co/alexandrainst/da-sentiment-base
[8]https://huggingface.co/vesteinn/danish_sentiment

| Lexicon/BERT Model | Valid. | Test |
|---|---|---|
| **Lexicon-based** | | |
| Sentida | 0.64 | 0.63 |
| TextBlob | 0.56 | 0.52 |
| VADER | 0.59 | 0.62 |
| **Supervised (without fine-tuning)** | | |
| Danish BERT Tone | 0.59 | 0.62 |
| Danish Sentiment | 0.71 | 0.74 |
| **Supervised (with fine-tuning)** | | |
| Danish BERT BotXO | 0.50 | 0.52 |
| Danish BERT Tone | 0.59 | 0.70 |
| Danish Sentiment | 0.63 | 0.72 |

Table 3: Lexicon-based and supervised sentiment classification F1-Score using different methods on validation and testing sets.

has been fine-tuned on 198M tweets specifically for sentiment analysis.

## 5.2 Lexicon-based classification experiment

Lexicon-based sentiment analysis is a technique used to determine the sentiment of a given text by assigning positive, negative, or neutral values to individual words based on their meanings. This approach relies on a pre-built sentiment lexicon, which contains a list of words and their corresponding sentiment values. The sentiment of a text is then calculated by summing the sentiment values of the individual words within the text. We evaluate the performance of our sentiment classifiers on the validation and testing split of the sub-corpus dataset on the Sentida, TextBlob, and VADER lexicons.

The Sentida classifier achieves the best results with 64% and 63% on the validation and testing sets, respectively. The results of the other two lexicons are presented in Table 3. Further analysis reveals that Sentida is the best at transferring to the unseen domain and language variation of historical literary Danish and Norwegian.

## 5.3 Supervised classification experiments (without fine-tuning)

To predict sentiment from a pre-trained model without fine-tuning, we first load the pre-trained model, format the input data and make predictions. However, without fine-tuning the model to rec-

ognize patterns specific to a particular sentiment analysis task, its performance may be limited.

We evaluate the performance of two pre-trained Danish BERT models, the Danish BERT Tone, and Danish Sentiment. The results show that the Danish Sentiment overpasses the Danish BERT Tone in both, evaluation and testing sets.

### 5.4 Supervised classification experiments (with fine-tuning)

The dataset used in lexicon-based and supervised without fine-tuning classification is also utilized in conducting supervised classification experiments. We fine-tune and evaluate three different pre-trained language models (BERT BotXO, Danish BERT Tone, and Danish Sentiment) in our dataset.

In this experiment, we fine-tune the described pre-trained BERT models on a task-specific dataset for sentiment classification. The dataset consists of 2,748 labeled sentences, with an almost balanced distribution of positive, neutral, and negative sentiments. We use a batch size of 32 and train the model for 30 epochs, using the AdamW optimizer with a learning rate of $10^{-3}$ (Loshchilov and Hutter, 2017). We evaluate the performance of each model using F1-score. Here we observe larger differences between validation and testing, which is a result of the fact that model selection (number of training epochs) was performed using the validation set. The Danish Sentiment model achieved the highest F1-score of 63% and 72% on the validation and test sets, respectively. Table 3 shows details about the obtained results from each model.

## 6 Classifier-assisted Corpus Analysis

We employ the 'Danish Sentiment' BERT model, which has shown to be the top-performing model, for predicting the sentiment of all sentences in the main corpus. We align the sentiment with the author's gender and the novel's year of publication. In Figure 1, the distribution of sentiment levels is depicted in relation to the author's gender and the percentage of sentences. Notably, female authors tend to exhibit a higher proportion of both negative and positive sentiments compared to male authors, on average.

These results provide insights into potential differences in sentiment expression between male and female authors in the analyzed data. The literary hypothesis about female authorships being

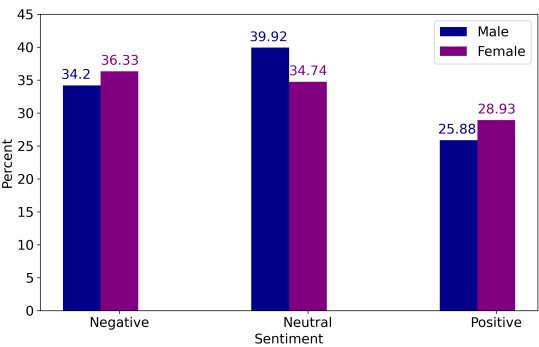

Figure 1: Distribution of sentiment and the author's genders. The X-axis is the sentiment class. Y-axis is the percentage of sentences per sentiment category.

'unhappy' is partly confirmed by our analysis. The female authors did express more negative sentiment, but also more positive sentiment than the male authors. Thus the implied positivity of the male authors, which is also part of the literary hypothesis, is not confirmed. A preliminary new hypothesis could therefore be that the female authors of the period wrote with a more expressed sentiment, whilst the male authors had a tendency towards a more unaffective style of writing.

Figure 2 provides a detailed analysis of the sentiment distribution over time concerning the author's gender. This figure shows the changes in sentiment tendencies across different gender groups and how these sentiments evolve over a given period. The results provide valuable insights into the dynamic nature of sentiments based on gender and can aid in understanding the evolving nature of sentiments and gender influences. The figure demonstrates that female authors exhibit more negative and positive sentiments over time than male authors.

Additionally, we calculated the overall sentiment by using a weighted average approach that considered both the sentiment distribution and weights of positive, neutral, and negative sentiments. Specifically, we assigned a weight of 3 to negative sentiment, 2 to neutral sentiment, and 1 to positive sentiment. To compare the sentiment of male and female authors over time, we computed the average sentiment scores over a period of 29 years and plotted them in Figure 3. Higher scores indicated more negative sentiment, while lower scores indicated more neutral or positive sentiment. The results in the figure show that, on

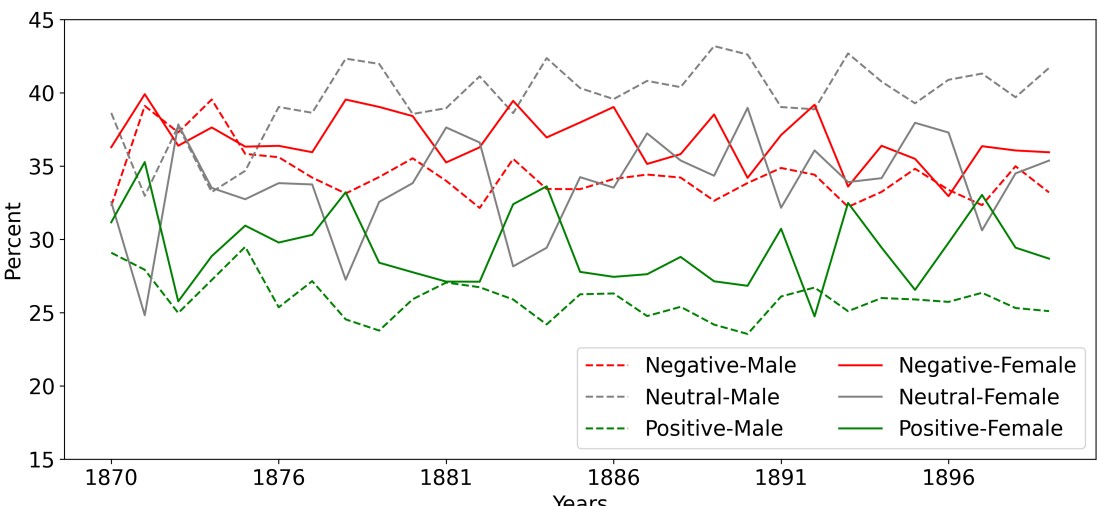

Figure 2: Distribution of sentiment over time. The X-axis is the years. The Y-axis is the percentage of sentences of a particular sentiment out of all sentences by authors of the same gender from that year.

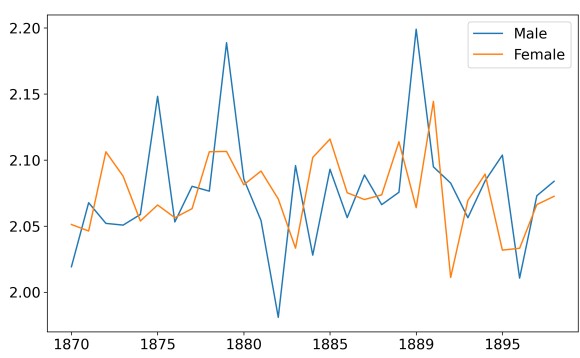

Figure 3: Average sentiment for both male and female authors over time.

| Sentiment Class | p-value |
|---|---|
| Negative | $5.20 \times 10^{-6}$ |
| Neutral | $5.13 \times 10^{-8}$ |
| Positive | $4.57 \times 10^{-10}$ |

Table 4: Significance of sentiment class differences: p-values from t-tests comparing mean male and female sentiment groups over the years, revealing statistically significant differences in sentiment trends.

average, the negative sentiment scores of female authors are higher than those of male authors in 16 out of the 29 years analyzed.

The significance of our findings was assessed by performing a t-test for each sentiment category, with the null hypothesis that the mean number of sentences with that category across the years is the same for sentences by male and female authors. In statistical analysis, a p-value lower than 0.05 is generally considered statistically significant and enables us to reject the null hypothesis, indicating the counts are statistically different. Table 4 shows small, statistically significant p-values for all sentiment classes.

## 7  Discussion

These initial analyses of a literary corpus with the newly developed sentiment classifier both confirm and contest the thesis of the 'unhappy' female texts. On the face of it, it seems that the female authors in the corpus express not only negative but also positive sentiments. At the same time, the second (implicit) part of the thesis is refuted, as the male authors do not write more positively. Clearly, this is reopening the literary discussion of the unhappy text with a level of complexity that these results cannot account for alone.

Further work is needed. In terms of computational interventions, a more in-depth analysis would call for us to investigate polarity in greater detail (how positive or negative are the novels?), while also paying more attention to narrative structures (how does polarity relate to

plot?).

More trivial analytical next steps involve employing segmentation of data and comparisons of sub-corpora. Are there, for example, certain works or authors that are outliers in terms of gender or valence? Are there any correlations when comparing canonized works and popular literature? By zooming in on either part of the novels (the endings, for example) or parts of the corpus (with regards to time slots or subgroups of producers/individual authors), specific strong trends may stand out.

Also, by working with smaller subsets of the corpus, we could perhaps challenge the essentialist idea of an 'Écriture Féminine', a unique feminine style of writing. Indeed, instead of simply asking whether there is a difference between male and female authorship in the period, we would also be able to explore how this difference is created in the corpus and possibly cultural, historical, gendered, and narratological reasons for it. In this connection, the fact that male authors seemingly write more 'neutrally' while female authors write with a higher degree of valence might be put into perspective through Sara Ahmed's theory of the 'stickiness' of emotions. Inspired by performativity theory, Sara Ahmed is interested in what emotions or affects do, rather than what they are (Ahmed, 2004). Both 'positive' affects, such as (naïve) joy, and 'negative' affects, such as shame have, especially during the period in question but also still today, clung to the feminine, while hegemonic masculinity has been and is associated with the rational and unsentimental (Connell, 1995).

In other words, a further refinement of the analytical steps goes hand in hand with critical interactions with the theoretical framework and specific historical contexts concerning the concepts of gender and emotion.

## 8   Conclusion

In this work, we used the MEMO corpus to create a high-quality human-annotated sentiment dataset for historical literary Danish and Norwegian. Despite multiple challenges, we showed that the task is feasible and that inter-annotator agreement is sufficiently high to warrant the use of the dataset for the training and evaluation of sentiment classifiers. In an extensive evaluation of such models, we found that the best performance is obtained with XLM-RoBERTa fine-tuned on senti-

ment analysis of modern Danish tweets and then further on the training set from our dataset. Using this model to annotate the whole MEMO corpus automatically, we observe that, as the literary theory predicts, female authors expressed more negative sentiments than male authors in the period. However, the hypothesis of the unhappy texts is only partially confirmed by our analysis as the numbers also suggest that the female authors, in contrary to the hypothesis, expressed more positive sentiments than the male authors who collectively expressed more neutral sentiments than their female counterparts.

In future work, we intend to experiment with further improvements of the classifier, including pre-training a language model on the whole corpus as a basis for fine-tuning the sentiment dataset. Furthermore, we intend to complement the classifier with topic models to investigate the evolution of sentiment towards specific topics (cultural change) over time and the evolution of words used to express a sentiment (language change).

From a literary perspective, a sentiment classifier trained on Scandinavian novels from the end of the 19$^{th}$ century holds great potential outside of the context of our specific test case. To give just one example, it would be interesting to study the so-called 'naturalism' from the time of the Modern Breakthrough. This literary movement or current is often said to be characterized by a pronounced pessimistic worldview. One can assume that negative segments predominate naturalist writings, and if that is in fact the case, the numbers would be interesting to compare with the rest of the literature from the period, i.e., canonical, realistic novels and this period's many forgotten texts.

From a literary theoretical perspective, a classifier such as this can be used to shed light on the importance of narratological layers at a macro level. Although the human reader is far superior to algorithms in separating character, narrator, and (implied) author, the numerical output of the computer is very interesting for existing research in the period. Questions like 'Does the sentiment distribution correspond to the overall assertions of the individual works?' and 'How does sentimental quantity relate to literary expression?' become possible to investigate.

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
