# OpenReview forum: "Sentiment Classification of Historical Danish and Norwegian Literary Texts"
_NoDaLiDa/2023/Conference — NoDaLiDa 2023_

### Official Review · Reviewer_k16R · 2023-02-22
**A study of sentiment in historical literary texts**

**Rating:** 7
**Confidence:** 4

**Review:**

The present paper describes analysis of sentiment in historical literary texts focusing on detecting negative, positive and neutral sentences (or segments?) in literary texts produced by either female or male authors in the 19th century. The central hypothesis of the study is that writing of female authors was characterised by negative sentiment, what the authors call 'unhappy texts' following existing literary studies in the field of female literature of this time period. In Section 2, the authors introduce the development of literature of this time with a focus on female authors becoming recognised. However, it remains unclear why texts by female authors are regarded to be 'unhappy'. This should be clarified so that the reader do not have to search for the cited works that claim so.
Another issue that should also be clarified in the paper is the definition of sentiment by the authors. They mention emotions and polarity but do not clearly state what they focus on in the presented analysis.

Annotation process also needs more clarification. For instance, were the annotators native speakers of what? Were they aware if the segment they evaluate was produced by a female or male author? Are there any published annotation guidelines the authors can refer to? What was the size of the annotated segments? The information provided is not enough. Why is the IAA moderate - the authors should add at least a comment on the reasons which are probably related to the challenges. If IAA is not so good, how reliable are then the results?

The description of data does not mention the distribution b/n Danish and Norwegian literature. Then, all the resources used were actually for Danish. Is there any adaptation for Norwegian needed? I think that the paper needs a clarification on this - having Danish language only in Danish and Norwegian literature of the 19th century?

In the presentation of the results some more interpretations are desirable. For instance, why is the difference between validation and test for Danish sentiment presented in Table 3 much bigger than in case of other tools and scenarios? Add more comments on the results presented in Table 3.
Texblob contains back-translations. Are the authors aware of changes that translation may bring to the sentiment (see e.g. work by Troiano et al. 2020 on changes in sentiment through translation).
For figure 1,  please also comment on the fact that female writers seem to be less neutral. Maybe it is not about being more negative than male authors but about being more expressive, containing more emotions and sentiment than works by male authors. But maybe this is just due to the selection of data? Some segments in a book could contain more emotions than the other  add more comments also on the data selection. Should the numbers in Figure 1 add to 100%?

665-667: 'These results provide insights' - which insights? What did we learn?
718-719: I am not sure if I agree on that the hypothesis was confirmed. Segments by female authors were also more positive than those by male authors. Why should they be then unhappy?

Check the reference list, some citation are not capitilised but should be, e.g. title in Shmidt&Bughardt.

105: reference to Andersen?
230: what is GDP?
581: what does 'that' in 'that is due' refer to?

References
Enrica Troiano, Roman Klinger, and Sebastian Padó. 2020. Lost in Back-Translation: Emotion Preservation in Neural Machine Translation. In Proceedings of the 28th International Conference on Computational Linguistics, pages 4340–4354, Barcelona, Spain (Online). International Committee on Computational Linguistics.



**Paper Type:**

Long paper

---

### Official Review · Reviewer_52vE · 2023-03-07
**The work contains flaws, the sentiment analysis was performed with an extremely small dataset (of which texts were taken from historical novels).**

**Rating:** 3
**Confidence:** 4

**Review:**

The work contains flaws, it is performed with an extremely small dataset. My remarks:
1.	You randomly select sentences and annotate them despite their context. What are you going to do with the models trained on such data? What is the relevance of your work?
2.	In line 293 you write that 2700 sentences were selected. From lines 305-306, it seems there were 2706 (1029 + 1029 + 648). In Table 2, there are 2705. In your experiments (lines 467-470) there are 2748 (2376+272+100) samples in total.
3.	What is the distribution among Danish and Norwegian historic texts in your sentiment dataset?
4.	If I understand correctly, different annotators annotated different texts (1029 + 1029 + 648) and there was no overlapping. How can you calculate Kappa then (line 454)? Please, clarify.
5.	As the methods (rule-based and dictionary-based approaches are not models) (Section 5.1) you select rule/dictionary-based approaches and supervised ML techniques. In the Introduction section, you explain that historic texts require specific solutions; however, you are using methods that are adjusted to deal with contemporary texts.
6.	Sentida, TextBlob, and VADER are NOT zero-shot methods (please, find more explanation here: https://huggingface.co/tasks/zero-shot-classification): they are lexicon/rule-based approaches. You are not performing any zero-shot learning in your research.
7.	Can “Danish” models (lines 521, 528, 662) be also effective for the Norwegian historic texts? Why?
8.	 In lines 607-610 you say that your dataset consists of 2705 labeled sentences with a balanced distribution among positive/negative/neutral classes. How is that possible? I see in Table 2 that your dataset is not balanced.
9.	In lines 607-613 you provide the list of hyper-parameter values. How were these values where selected, are they optimized?
10.	Can the distribution of male/female positive/negative/neutral sentiments even be considered: you have randomly selected sentences from novels. Does it somehow reflect the larger picture?
11.	From all tested methods XLM-RoBERTa-base is the most powerful transformer model (from all you tested). It is trained and fine-tuned on contemporary texts. Since your training dataset is extremely small (~2700 texts for training, validating, and testing) XLM-RoBERTa-base model is effective probably because it can find contemporary words in the texts you have in your dataset.

**Paper Type:**

Long paper

---

### Official Review · Reviewer_77Ve · 2023-03-10
**A presentation of a valuable corpus along with some preliminary experiments**

**Rating:** 7
**Confidence:** 3

**Review:**

The paper presents interesting work and is well written. The corpus presented seems valuable, and while the findings from the experiments are unsurprising, they seem to merit addition to the research literature.

I believe the paper should be accepted for publication. I have no major objections, but some minor comments follow below.

- I would advise against highlighting the references with colored squares, as this impedes readability.

- The hyphenation is a little excessive. I would recommend raising the hyphenpenalty in the Latex code.

- 081: "(see §2)" is odd to put immediately before §2. "outlined in the following section" would flow better.

- 092: historio\\-graphy  (to avoid the weird hyphenation here)

- 111: Move reference: "_Hovedsporet_ (Jørgensen, 2005)"

- 206: nowadays is on

- 210: New paragraph before "Several"?

- 254: Why is "(§5)" put here?

- 289: "Our main corpus currently comprises 900 novels from 1870-1900, specifically Danish and Norwegian historical novels." This does not seem to be in agreement with Table 1, which puts the number of novels at 839.

- 303: I would put "PhD" rather than "graduate", as a master student is also a graduate student.

- 310: I think "vested" is probably not the right word here.

- 340: Is the apostrophe after "saa" intended?

- 409: inexpressedtion

- 415: Tranl ion

- 462–477: What about the remaining 7% of the dataset?

- 476 and throughout: "F1 score" is more standard than "F1-score" (even though it is an "F-score")

- 559: You might make it even more explicit that this model is a fine-tuning of the previous one.

- 565: fine-tuned ~~using~~ specifically

- 577: achieves the best results, with F1 scores of 64%…

- 580: Unclear what "this" refers to.

- 612: 0.001 (or $10^{-3}$) rather than 1e-3

- 632–642: This paragraph seems somewhat unnecessary

- 643–667: … and this one seems to be a rehashing of it.

- 708: This caption is a little unclear.
- 708: ~~Comparison of~~
- 708: t-tests

- 756: Figure 2 is hard to read. Try using the same colors for negative, neutral and positive across genders, and differentiate the two genders with solid vs. dashed lines.

- 784: the punctuation needs to change here. I would suggest "have, especially in the period but still today, clung to…"

- 847: if that is in fact the case

- 864–1010: References are full of erroneous lower-case. 869: c, p ,e. 879: q. 885: ocr. 925: l. 953: d. 962: dx2. 974: g, e, l. 984: l, s, d, a. 1007: g. There might be more.

- 887: Stray ellipsis ("...") ? Title should be _Det moderne gennembruds mænd_.

- 953: hyphen should be dash

**Paper Type:**

Long paper

---

### Decision · Program_Chairs · 2023-03-17

Accept